# LoftQ: LoRA-Fine-Tuning-Aware Quantization for Large Language Models

**Yixiao Li**[1] *  **Yifan Yu**[1] *  **Chen Liang**[1]  **Pengcheng He**[2]

**Nikos Karampatziakis**[2]  **Weizhu Chen**[2]  **Tuo Zhao**[1]

## ABSTRACT

Quantization is an indispensable technique for serving Large Language Models (LLMs) and has recently found its way into LoRA fine-tuning (Dettmers et al., 2023). In this work we focus on the scenario where quantization and LoRA fine-tuning are applied together on a pre-trained model. In such cases it is common to observe a consistent gap in the performance on downstream tasks between full fine-tuning and quantization plus LoRA fine-tuning approach. In response, we propose LoftQ (**Lo**RA-**F**ine-**T**uning-aware **Q**uantization), a novel quantization framework that simultaneously quantizes an LLM and finds a proper low-rank initialization for LoRA fine-tuning. Such an initialization alleviates the discrepancy between the quantized and full-precision model and significantly improves generalization in downstream tasks. We evaluate our method on natural language understanding, question answering, summarization, and natural language generation tasks. Experiments show that our method is highly effective and outperforms existing quantization methods, especially in the challenging 2-bit and 2/4-bit mixed precision regimes. The code is available on https://github.com/yxli2123/LoftQ.[1] [2]

## 1 INTRODUCTION

The advent of Pre-trained Language Models (PLMs) has marked a transformative shift in the field of Natural Language Processing (NLP), offering versatile solutions across various applications (He et al., 2021b; Lewis et al., 2019; Touvron et al., 2023). They have showcased unparalleled proficiency in executing a variety of language tasks, including Natural Language Understanding (NLU) and Natural Language Generation (NLG). These models typically have millions or even billions of parameters, necessitating substantial computational and memory requirements. However, the extensive computational and memory demands of these models pose significant challenges, especially for deployments where resources are often constrained and need to be shared among many users.

To mitigate the extensive storage requirements of pre-trained models, quantization serves as a pivotal compression technique (Zafrir et al., 2019; Shen et al., 2020; Bai et al., 2022; Dettmers et al., 2022), converting high-precision numerical values into a discrete set of values. Typically, model parameters, originally stored in a 16-bit float format, are transformed into a 4-bit integer format through quantization, resulting in a substantial 75% reduction in storage overhead. Additionally, to facilitate the adaptation of quantized pre-trained models to downstream tasks efficiently, Low-Rank Adaptation (LoRA) is a viable approach (Hu et al., 2021). This technique is a parameter-efficient fine-tuning method traditionally applied to high-precision pre-trained models. It is based on the hypothesis that the differences between fully fine-tuned weights and pre-trained weights exhibit low-rank properties. This allows these differences to be represented using low-rank matrices. As a result, the original pre-trained weights remain unaltered, with adaptations confined solely to these low-rank matrices, enabling effective task adaptation.

---

*Equal contribution

[1]Li, Yu, Liang and Zhao are affiliated with Georgia Institute of Technology. Correspondence to yixiaoli@gatech.edu, yyu429@gatech.edu and tourzhao@gatech.edu.

[2]He, Karampatziakisand and Chen are affiliated with Microsoft Azure.

When quantizing pre-trained models, practitioners often concentrate primarily on the quantization technique, inadvertently neglecting the importance of subsequent LoRA fine-tuning (Dettmers et al., 2023; Diao et al., 2023). For example, QLoRA inherits the fixup initialization (Zhang et al., 2019) used in LoRA, which (Dettmers et al., 2023) attaches zero initialized low-rank adapters (see Section 2.3) to the quantized pre-trained model. The inevitable discrepancy introduced by quantization during the approximation of the original high-precision numbers, a scenario particularly pronounced in low-bit situations such as the 2-bit regime, can adversely impact the initialization of LoRA fine-tuning. As illustrated in Figure 1a, the quantized pre-trained model obtained by QLoRA exhibits severe degradation below the 3-bit level. This deviation in initialization often results in an inferior fine-tuning performance. As illustrated in Figure 1b, the fine-tuning performance drops as the quantization bit decreases when applying QLoRA. Moreover, it is noteworthy that QLoRA fails below the 3-bit level.

In this paper, we introduce a novel quantization framework, called **Lo**RA-**F**ine-**T**uning-aware **Q**uantization (LoftQ). It is designed specifically for pre-trained models that require quantization and LoRA fine-tuning. This framework actively integrates low-rank approximation, working in tandem with quantization to jointly approximate the original high-precision pre-trained weights. This synergy significantly enhances alignment with the original pre-trained weights as illustrated in Figure 2. Consequently, our method provides an advantageous initialization point for subsequent LoRA fine-tuning, leading to improvements in downstream tasks.

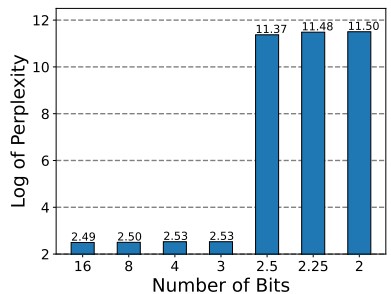
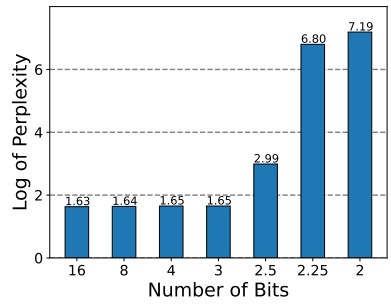

(a) Pre-trained LLAMA-2-13b on WikiText-2     (b) Fine-tuned LLAMA-2-13b on WikiText-2

Figure 1: QLoRA performance with different bits. **Left:** QLoRA initialization of LLAMA-2-13b on WikiText-2. **Right:** Apply QLoRA to LLAMA-2-13b on WikiText-2 language modeling task. Smaller perplexity indicates better performance.

We evaluate our quantization framework by conducting extensive experiments on downstream tasks, such as NLU, question answering, summarization, and NLG. Experiments show that LoftQ consistently outperforms QLoRA across all precision levels. For instance, with 4-bit quantization, we achieve a 1.1 and 0.8 gain in Rouge-1 for XSum (Narayan et al., 2018) and CNN/DailyMail (Hermann et al., 2015), respectively. LoftQ excels particularly in low-bit scenarios and works effectively with different quantization methods. For example, we achieve over an 8% gain on MNLI (Wang et al., 2019) and more than 10% on SQuADv1.1 (Rajpurkar et al., 2016) with both 2-bit NormalFloat and the 2-bit uniform quantization. We have not seen our approach performs worse than QLoRA.

## 2 BACKGROUND

### 2.1 TRANSFORMER MODELS

A transformer model contains a sequence of layers, where each layer consists of two sub-layers: a multi-head self-attention (MHA) and a fully connected feed forward network (FFN) (Vaswani et al., 2017). Given the input $X \in \mathbb{R}^{n \times d}$, where $n$ is the sequence length and $d$ is the hidden dimension of the model, MHA computes the $h$ attention heads in parallel:

$$\text{MHA}(X) = \text{Concat}(\text{head}_1, ..., \text{head}_h)W_o,$$

$$\text{where} \quad \text{head}_i = \text{Softmax}(XW_{q_i}(XW_{k_i})^\top / \sqrt{d_h})XW_{v_i} \quad \text{for} \quad i = 1, ..., h,$$

where $W_{q_i}, W_{k_i}, W_{v_i} \in \mathbb{R}^{d \times d_h}$ are query, key, and value matrices, $W_o \in \mathbb{R}^{d \times d}$ is the output matrix, and $d_h = d/h$. FFN comprises two linear transformations and an activation function, and is defined as $\text{FFN}(X) = \sigma(XW_{f_1} + b_1)W_{f_2} + b_2$, where $W_{f_1} \in \mathbb{R}^{d \times d_m}$, $W_{f_2} \in \mathbb{R}^{d_m \times d}$, and $\sigma(\cdot)$ is the activation function. A residual connection is used and followed by layer normalization.

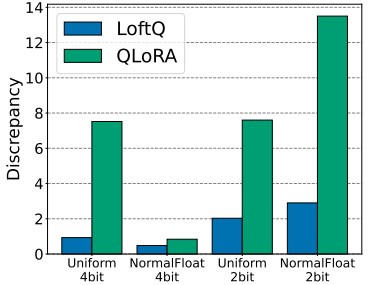
(a) Spectral norm of the initialization difference

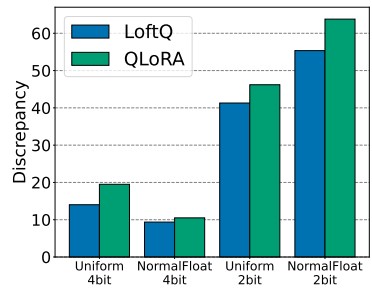
(b) Frobenius norm of the initialization difference

Figure 2: Initialization discrepancy between the LoRA initialization and the original pre-trained weight matrix, described by the spectral norm and Frobenius norm of the difference. The weight matrix in the above figures is randomly selected in BART-large. The initialization is obtained by QLoRA and LoftQ, with Uniform and NormalFloat quantization methods applied at both 2-bit and 4-bit levels. LoftQ successfully mitigates the discrepancy, especially at the 2-bit level.

## 2.2 QUANTIZATION

**Quantization.** Given a high-precision number, e.g., such as 32-bit floating point number, $X^{\mathrm{HP}} \in \mathbb{R}$, $N$-bit quantization encodes it to an integer $X^{\mathrm{INT}} \in \{0, 1, ..., 2^N - 1\}$. This process can be expressed as

$$X^{\mathrm{INT}} = \mathrm{round}\left((2^N - 1)F\left(X^{\mathrm{HP}}\right)\right), \tag{1}$$

where $F(\cdot) \colon \mathbb{R} \mapsto [0, 1]$ is a normalization function. Uniform quantization assumes $F(X) = (X - X_{\min})/(X_{\max} - X_{\min})$. Dettmers et al. (2023) proposes 4-bit NormalFloat Quantization (NF4). It assumes $X \sim \mathcal{N}(0, \sigma^2)$ and hence $F(X) = \Phi(X/\sigma)$, where $\Phi(\cdot)$ is the cumulative distribution function of the standard normal distribution.

**Dequantization.** A lookup table $\mathcal{T}$, where

$$\mathcal{T}[i] = F^{-1}\left(\frac{i}{2^N - 1}\right), i = 0, 1, ..., 2^N - 1, \tag{2}$$

is used to decode the integer $X^{\mathrm{INT}}$ to its simulated high-precision counterpart $X^{\mathrm{D}} \in \mathbb{R}$. Therefore, the dequantization can be expressed as

$$X^{\mathrm{D}} = \mathcal{T}[X^{\mathrm{INT}}]. \tag{3}$$

**Simulated Quantization for Matrices.** While it is possible to perform multiplication directly between quantized representations, it is common to apply simulated quantization for matrices (Bai et al., 2020; Shen et al., 2020). There, quantized weight matrices are stored as encoded integers in memory, and are temporarily dequantized to simulated high-precision matrices by the lookup table when engaged in multiplication operations. In simulated quantization, it is only necessary to analyze the map from a high-precision matrix to a simulated high-precision matrix. We denote this end-to-end process by $q_N(\cdot) \colon \mathbb{R}^{m \times n} \mapsto \mathbb{R}_N^{m \times n}$, where $\mathbb{R}_N : \{\mathcal{T}[i] \in \mathbb{R} | 0 \leq i < 2^N\}$.

## 2.3 LOW-RANK ADAPTATION

LoRA (Hu et al., 2021) updates two small weight matrices $A$ and $B$ that are attached to a frozen pre-trained weight matrix $W$. Hence, a linear transformation, $Y = XW$, is reformulated as

$$Y = XW + XAB^\top, \tag{4}$$

where $X \in \mathbb{R}^{n \times d_1}, W \in \mathbb{R}^{d_1 \times d_2}, A \in \mathbb{R}^{d_1 \times r}, B \in \mathbb{R}^{d_2 \times r}$, and $r \ll \min\{d_1, d_2\}$. Initially,

$$A \sim \mathcal{N}(0, \sigma^2), \ B = 0, \tag{5}$$

so as to align to the pre-trained weights. During the fine-tuning, $W$ is fixed while $A$ and $B$ are updated by some SGD-type optimization method.

It is worth noting that if low-rank adapters $A$ and $B$ are attached to a quantized backbone $Q = q_N(W)$ and are initialized by (5), the starting weight $Q + AB^\top$ is no longer equal to the pre-trained weight $W$ due to the discrepancy introduced by the quantization.

## 3   METHOD

We propose **LoRA-Fine-Tuning-aware Quantization** (LoftQ), a quantization framework for LLMs. It alternatively applies quantization and low-rank approximation to approximate original pre-trained weights. This quantization framework provides a promising initialization for LoRA fine-tuning, which alleviates the quantization discrepancy in QLoRA and improves generalization in downstream tasks significantly.

### 3.1   LoRA-AWARE QUANTIZATION

We use an $N$-bit quantized weight $Q \in \mathbb{R}_N^{d_1 \times d_2}$ and low-rank approximations $A \in \mathbb{R}^{d_1 \times r}, B \in \mathbb{R}^{d_2 \times r}$ to approximate the original high-precision pre-trained weight $W \in \mathbb{R}^{d_1 \times d_2}$ as the initialization of LoRA fine-tuning. Specifically, before fine-tuning, we initialize the network by minimizing the following objective:

$$\min_{Q,A,B} \left\| W - Q - AB^\top \right\|_F,  \tag{6}$$

where $\|\cdot\|_F$ denotes the Frobenious norm. This objective in (6) takes LoRA fine-tuning into consideration by jointly optimizing the initial values of the quantized backbone $Q$ and low-rank adapters $A, B$. Contrarily, practitioners typically convert the pre-trained weight $W$ into a quantized weight $Q$ outright, neglecting the subsequent LoRA fine-tuning process. This oversight leads to notable performance degradation in downstream tasks arising from the quantization discrepancy.

### 3.2   ALTERNATING OPTIMIZATION

We solve the minimization problem in (6) by alternating between quantization and singular value decomposition (SVD). To begin with, we set $A_0$, and $B_0$ equal to 0.

**Quantization**. At the $t$-th step, we quantize the difference between the original pre-trained weight matrix $W$ and the low-rank approximation $A_{t-1}B_{t-1}^\top$ from the previous step to obtain the quantized weight matrix $Q_t$ by

$$Q_t = q_N(W - A_{t-1}B_{t-1}^\top),  \tag{7}$$

where $q_N(\cdot)$ maps a high-precision weight matrix to a quantized matrix.

We remark that our algorithm is compatible with different quantization functions $q_N(\cdot)$. We apply NF4 and the uniform quantization in Section 4 as examples. We also remark that $Q_t$ is not an exact solution of the minimization in (6), given the fixed $A_{t-1}B_{t-1}^\top$, but it is an efficient approximation.

**SVD**. After obtaining the $t$-th quantized weight $Q_t$, SVD is applied to the residual of the quantization denoted by $R_t = W - Q_t$ by

$$R_t = \sum_{i=1}^{d} \sigma_{t,i} u_{t,i} v_{t,i}^\top,  \tag{8}$$

where $d = \min\{d_1, d_2\}$, $\sigma_{t,1} \geq \sigma_{t,2} \geq ... \geq \sigma_{t,d}$ are the singular values of $R_t$, $u_{t,i}$'s and $v_{t,i}$'s are the associated left and right singular vectors of $R_t$. We then obtain a rank-$r$ approximation of $R_t$ by $A_t B_t^\top$, where

$$\begin{aligned} A_t &= [\sqrt{\sigma_{t,1}}u_{t,1}, ..., \sqrt{\sigma_{t,r}}u_{t,r}], \\ B_t &= [\sqrt{\sigma_{t,1}}v_{t,1}, ..., \sqrt{\sigma_{t,r}}v_{t,r}]. \end{aligned}  \tag{9}$$

We summarize our method in Algorithm 1. It is worth noting that $T = 1$ is a special case where $Q_1$ is the exact quantized weight obtained by QLoRA, and low-rank approximations $A_1, B_1$ are obtained by the SVD of the quantization residual $W - Q_1$. $T = 1$ is sufficient to mitigate the quantization discrepancy, and alternating optimization helps to find a closer initialization to the pre-trained weight $W$, which further improves the performance (see Section 3).

We remark that the computational cost of LoftQ is negligible because it is applied to individual weight matrices and can be executed in parallel. We also remark one can apply LoftQ only once to a pre-trained model and reuse the initialization obtained by LoftQ for different downstream tasks.

### 3.3   APPLYING TO LoRA FINE-TUNING

We store the $Q_T \in \mathbb{R}_N^{d_1 \times d_2}$ obtained by LoftQ using an integer matrix $M$ by (1) and a lookup table $\mathcal{T}$ by (2). We initialize the backbone with the integer matrix $M$ and initialize the low-rank adapters with $A_T, B_T$ obtained by LoftQ.

---

**Algorithm 1** LoftQ

---

**input** Pre-trained weight $W$, target rank $r$, $N$-bit quantization function $q_N(\cdot)$, alternating step $T$
 1: Initialize $A_0 \leftarrow 0, B_0 \leftarrow 0$
 2: **for** t = 1 to $T$ **do**
 3:     Obtain quantized weight $Q_t \leftarrow q_N(W - A_{t-1}B_{t-1}^\top)$
 4:     Obtain low-rank approximation $A_t, B_t \leftarrow \text{SVD}(W - Q_t)$ by (9)
 5: **end for**
**output** $Q_T, A_T, B_T$

---

During LoRA fine-tuning, we freeze the integer weight $M$ and optimize the low-rank adapters with an efficient optimization algorithm, e.g., AdamW (Loshchilov & Hutter, 2017). In forward propagation, the integer weight $M$ is temporarily dequantized to the simulated high-precision weight $Q_T$ by its lookup table, as described in (3). In back propagation, gradients and optimizer state are only related to low-rank adapters $A, B$, which reduces considerable training cost.

## 4  EXPERIMENTS

We evaluate our method on NLU and NLG tasks. We apply LoftQ for quantizing DeBERTaV3-base (He et al., 2021b), BART-large (Lewis et al., 2019), and LLAMA-2 series (Touvron et al., 2023).

**Implementation Details.** Following the prior works of LoRA variants (Zhang et al., 2023; He et al., 2021a), we freeze all the backbone weight matrices and add low-rank adapters to weight matrices in MHA and FFN of all layers. We quantize the weight matrices that are attached by low-rank adapters. All the quantized models and adapters used in this paper are available on `https://huggingface.co/LoftQ`. Our implementation is based on publicly available *Huggingface Transformers* code-base (Paszke et al., 2019). All the experiments are conducted on NVIDIA A100 GPUs.

**Quantization Methods.** We apply two quantization methods to demonstrate LoftQ is compatible with different quantization functions:

- *Uniform quantization* is a classic quantization method. It uniformly divides a continuous interval into $2^N$ categories and stores a local maximum absolute value for dequantization.

- *NF4* and its 2-bit variant *NF2* are quantization methods used in QLoRA (Dettmers et al., 2023). They assume that the high-precision values are drawn from a Gaussian distribution and map these values to discrete slots that have equal probability.

We perform 2-bit and 4-bit quantization on all models, achieving compression ratios of 25-30% and 15-20% at the 4-bit and 2-bit levels, respectively. The compression ratios and trainable parameter ratios for all models are detailed in the Appendix A.

**Baselines.** We compare LoftQ with the following baseline methods:

- *Full fine-tuning* is the most common approach for adapting a pre-trained model to downstream tasks. The model is initialized with pre-trained weights and all parameters are updated through an SGD-type optimization method.

- *Full precision LoRA (LoRA)* is a lightweight method for task adaptation, where it stores the backbone using 16-bit numbers and optimizes the low-rank adaptors only. The adaptors are applied to the same matrices as in LoftQ.

- *QLoRA* is similar to *LoRA* except the backbone is quantized into low-bit regime. The low-rank adapters are initialized using (5) and are applied to the same matrices as in LoftQ.

### 4.1  ENCODER-ONLY MODEL: DEBERTAV3

**Models and Datasets.** We quantize the DeBERTaV3-base (He et al., 2021b) with LoftQ, then fine-tune and evaluate the model on the General Language Understanding Evaluation (GLUE) benchmark (Wang et al., 2019), SQuADv1.1 (Rajpurkar et al., 2016), and ANLI (Nie et al., 2019). The specific tasks of GLUE are given in Appendix C. Following previous works (Zhang et al., 2023), we exclude WNLI in the experiments.

**Implementation Details.** We select the learning rates from $\{1 \times 10^{-5}, 5 \times 10^{-5}, 1 \times 10^{-4}\, 5 \times 10^{-4}\}$. We quantize the entire backbone. Given that GLUE, SQuADv1.1, and ANLI are relatively easy NLU tasks, we also quantize the embedding layer for higher compression efficiency. We apply the NormalFloat and the uniform quantization for LoftQ and QLoRA at both 2-bit and 4-bit levels. We use rank 16 and 32 for low-rank adapters. More implementation details, such as the training epochs and batch sizes, are presented in Appendix D.2.

**Main Results.** Table 1 and Table 2 summarize the results for 2-bit quantization on the GLUE, SQuADv1.1, and ANLI datasets, by NF2 and the uniform quantization, respectively. Our method consistently outperforms QLoRA on all settings with respect to different ranks, quantization methods, and datasets. When using the uniform quantization (Table 2), our method achieves 88.0% accuracy on MNLI-m, surpassing the QLoRA baseline by 8%. For tasks like SST and SQuADv1.1, our method even approaches the full fine-tuning performance at 2-bit level. The 4-bit quantization experiment results are presented in Appendix D.1 as both LoftQ and QLoRA achieve performance close to full fine-tuning.

Table 1: Results with 2-bit LoftQ of DeBERTaV3-base models on GLUE development set, SQuADv1.1 development set, ANLI test set using **NF2 quantization**. We report the median over four seeds. *N.A.* indicates the model does not converge. The best results on each dataset are shown in **bold**.

| Rank | Method | MNLI
m / mm | QNLI
Acc | RTE
Acc | SST
Acc | MRPC
Acc | CoLA
Matt | QQP
Acc | STSB
P/S Corr | SQuAD
EM/F1 | ANLI
Acc |
|---|---|---|---|---|---|---|---|---|---|---|---|
| - | Full FT | 90.5/90.6 | 94.0 | 82.0 | 95.3 | 89.5/93.3 | 69.2 | 92.4/89.8 | 91.6/91.1 | 88.5/92.8 | 59.8 |
| 16 | LoRA | 90.4/90.5 | 94.6 | 85.1 | 95.1 | 89.9/93.6 | 69.9 | 92.0/89.4 | 91.7/91.1 | 87.3/93.1 | 60.2 |
| 16 | QLoRA
LoftQ | 75.4/75.6
**84.7/85.1** | 82.4
**86.6** | 55.9
**61.4** | 86.5
**90.2** | 73.8/82.8
**83.8/88.6** | N.A.
**37.4** | 86.8/82.3
**90.3/86.9** | 83.0/82.8
**87.1/86.9** | 61.5 / 71.2
**81.5/88.6** | N.A.
**47.1** |
| 32 | QLoRA
LoftQ | 78.5/78.7
**86.0/86.1** | 80.4
**89.9** | 56.7
**61.7** | 86.9
**92.0** | 73.8/82.7
**83.6/87.2** | N.A.
**47.5** | 87.1/82.7
**91.0/87.9** | 83.6/83.3
**87.5/87.0** | 64.6/73.8
**82.9/89.8** | N.A.
**49.0** |

Table 2: Results with 2-bit LoftQ of DeBERTaV3-base models on GLUE development set, SQuADv1.1 development set using **Uniform quantization** . We report the median over four seeds. *N.A.* indicates the model does not converge. The best results on each task are shown in **bold**.

| Rank | Method | MNLI
m / mm | QNLI
Acc | RTE
Acc | SST
Acc | MRPC
Acc | CoLA
Matt | QQP
Acc | STSB
P/S Corr | SQuAD
Em/F1 |
|---|---|---|---|---|---|---|---|---|---|---|
| - | Full FT | 90.5/90.6 | 94.0 | 82.0 | 95.3 | 89.5/93.3 | 69.2 | 92.4/89.8 | 91.6/91.1 | 88.5/92.8 |
| 16 | LoRA | 90.4/90.5 | 94.6 | 85.1 | 95.1 | 89.9/93.6 | 69.9 | 92.0/89.4 | 91.7/91.1 | 87.3/93.1 |
| 16 | QLoRA
LoftQ | 76.5/76.3
**87.3/87.1** | 83.8
**90.6** | 56.7
**61.1** | 86.6
**94.0** | 75.7/84.7
**87.0/90.6** | N.A.
**59.1** | 87.1/82.6
**90.9/88.0** | 83.5/83.4
**87.9/87.6** | 69.5/77.6
**84.4/91.2** |
| 32 | QLoRA
LoftQ | 79.9/79.5
**88.0/88.1** | 83.7
**92.2** | 57.8
**63.2** | 86.9
**94.7** | 76.5/84.5
**87.5/91.2** | N.A.
**60.5** | 88.6/84.7
**91.3/88.3** | 84.1/84.0
**89.5/89.2** | 71.6/80.2
**85.2/91.6** |

Our method is also more stable compared to QLoRA in the low-bit regime. For instance, while QLoRA fails to converge on CoLA for both quantization methods and ranks, LoftQ converges in all cases and achieves a score of 60.5 using uniform quantization at rank 32. LoftQ stands out in its ability to consistently attain robust and improved performance by effectively preserving the starting point of pre-trained weights.

## 4.2 ENCODER-DECODER MODEL: BART

**Models and Datasets.** We quantize BART-large model (Lewis et al., 2020) with LoftQ, then fine-tune and evaluate the model on two commonly used summarization datasets: XSum (Narayan et al., 2018) and CNN/DailyMail(Hermann et al., 2015).

**Implementation Details.** We apply LoftQ to weight matrices in MHA and FFN of both encoder and decoder layers. We report ROUGE 1/2/L scores, which are the metrics for summarization tasks (Lin, 2004). We conduct quantization experiments in both 2-bit and 4-bit scenarios. We experiment with both NormalFloat and the uniform quantization in both 2-bit and 4-bit scenarios. In each precision, we choose rank equal to 8 and 16 for a fair comparison with the full precision LoRA baseline (Zhang et al., 2023). Please see Appendix E for detailed configurations.

**Main Results.** Table 3 summarizes our 4-bit quantization experiment results on the XSum and CNN/DailyMail test sets. Our method consistently outperforms QLoRA at both ranks on both datasets. It even surpasses full precision LoRA at both ranks on Xsum. We will discuss this unexpected results in Section 5. The 2-bit quantization results are shown in Table 4. Our observation is consistent with the NLU experiments, that LoftQ demonstrates the convergence to reasonable results, while QLoRA does not converge. This indicates our method is robuster by narrowing the initialization gap.

Table 3: Results with 4-bit LoftQ of BART-large on XSum and CNN/DailyMail. We report ROUGE-1/2/L. *Lead-3* means choosing the first 3 sentences as the summary. *N.A.* indicates the model does not converge. *Full FT*: full fine-tuning. We report the median over five seeds.

| Quantization | Rank | Method | XSum | CNN/DailyMail |
|---|---|---|---|---|
| Full Precision | - | Lead-3
Full FT | 16.30/1.60/11.95
45.14/22.27/37.25 | 40.42/17.62/36.67
44.16/21.28/40.90 |
| | 8
16 | LoRA
LoRA | 43.40/20.20/35.20
43.95/20.72/35.68 | 44.72/21.58/41.84
45.03/21.84/42.15 |
| NF4 | 8 | QLoRA
LoftQ | 42.91/19.72/34.82
**44.08/20.72/35.89** | 43.10/20.22/40.06
**43.81/20.95/40.84** |
| | 16 | QLoRA
LoftQ | 43.29/20.05/35.15
**44.51/21.14/36.18** | 43.42/20.62/40.44
**43.96/21.06/40.96** |
| Uniform | 8 | QLoRA
LoftQ | 41.84/18.71/33.74
**43.86/20.51/35.69** | N.A.
**43.73/20.91/40.77** |
| | 16 | QLoRA
LoftQ | 42.45/19.36/34.38
**44.29/20.90/36.00** | 43.00/20.19/40.02
**43.87/20.99/40.92** |

Table 4: Results with 2-bit LoftQ of BART-large on XSum and CNN/DailyMail using **NF2 quantization**. *N.A.* indicates the model does not converge. We report ROUGE-1/2/L, the higher the better. We report the median over five seeds.

| Rank | Method | XSum | CNN/DailyMail |
|---|---|---|---|
| 8 | QLoRA
LoftQ | N.A.
39.63/16.65/31.62 | N.A.
42.24/19.44/29.04 |
| 16 | QLoRA
LoftQ | N.A.
40.81/17.85/32.80 | N.A.
42.52/19.81/39.51 |

### 4.3 DECODER-ONLY MODEL: LLAMA-2

**Models and Datasets.** We quantize LLAMA-2-7b and LLAMA-2-13b (Touvron et al., 2023) with LoftQ. We then fine-tune and evaluate the models on two NLG datasets: GSM8K (Cobbe et al., 2021) and WikiText-2 (Merity et al., 2016). Please see Appendix F for more details about the datasets.

**Implementation Details.** Similarly, we apply LoftQ to weight matrices in MHA and FFN of all layers. In WikiText-2 evaluation, we report perplexity. In case of overfitting, we apply weight decay to low-rank adapters for all settings. In GSM8K evaluation, we extract numerical answers in the generated solutions and then calculate the accuracy using those numerical answers. We conduct experiments with both NF2 and NF4. Please see Appendix F for detailed configurations.

**Main Results.** Table 5 presents a summary of our experiments on LLAMA-2-7b and LLAMA-2-13b using 2-bit, 4-bit, and mixed-precision NormalFloat quantization methods on WikiText-2 and GSM8K datasets. In WikiText-2, our method consistently outperforms QLoRA across all quantization precision settings on both models. When dealing with the challenging 2-bit precision, where QLoRA fails to converge, LoftQ manages to achieve a perplexity of 7.85. In GSM8K, our method achieves better or on par performance compared to QLoRA across different model sizes and quantization precision levels. For example, our method achieves 26.5% accuracy using 2-bit precision of LLAMA-2-7b, where QLoRA does not converge.

To provide a customized trade-off between the performance and precision, we also explore mixed-precision (equivalent to 3 bits) quantization where matrices in the first half layers are quantized using 4 bits, and the rest matrices remain 2 bits. We witness a remarkable 4.1% accuracy boost on the GSM8K dataset using LLAMA-2-7b and a 4.7% boost using LLAMA-2-13b. This result underscores the potential of LoftQ for complex mixed-precision quantization scenarios.

Table 5: Results of LoftQ using NormalFloat for LLAMA-2 series on WikiText-2 and GSM8K. 3/2.5/2.25-bit indicates mixed-precision quantization: 4-bit precision for the first 16/8/4 layers and 2-bit precision for the rest of layers. We report the perplexity (the smaller the better) for WikiText-2 and accuracy for GSM8K. The rank of low-rank adapters is 64. *N.A.* indicates the model does not converge. We report the median over five random seeds.

| Method | Bit | LLAMA-2-7b | | LLAMA-2-13b | |
|---|---|---|---|---|---|
| | | WikiText-2↓ | GSM8K↑ | WikiText-2↓ | GSM8K↑ |
| LoRA | 16 | 5.08 | 38.5 | 5.12 | 48.8 |
| QLoRA | 4 | 5.70 | **38.2** | 5.22 | 48.8 |
| LoftQ | 4 | **5.24** | 38.0 | **5.16** | **49.1** |
| QLoRA | 3 | 5.73 | 32.1 | 5.22 | 40.7 |
| LoftQ | 3 | **5.63** | **36.2** | **5.13** | **45.4** |
| QLoRA | 2.5 | N.A. | N.A. | 19.39 | N.A. |
| LoftQ | 2.5 | 5.78 | **31.1** | 5.22 | **41.1** |
| QLoRA | 2.25 | N.A. | N.A. | N.A. | N.A. |
| LoftQ | 2.25 | **6.13** | **27.5** | **5.45** | **38.1** |
| QLoRA | 2 | N.A | N.A. | N.A. | N.A. |
| LoftQ | 2 | **7.85** | **26.5** | **7.69** | **33.4** |

## 4.4 ANALYSIS

**Effectiveness of Alternating Optimization.** We conduct experiments with different alternating step $T$ to verify the effectiveness of the alternating optimization and to find the best value $T$ as a hyperparameter for different models. Across all tasks and models, we observed that alternating optimization yields substantial improvements even with a minimal alternating step. This suggests that it rapidly narrows the discrepancy between quantized weights and pre-trained weights, making our method easy to apply. For example, LoftQ achieves 21.14 Rouge-2 score on XSum using only 1 step. Interestingly, we noticed that increasing the alternating step beyond a certain point tends to result in diminishing returns. We suspect this phenomenon occurs because, as the gap becomes smaller, it becomes more challenging for alternating optimization to consistently minimize the gap at each step. This challenge emerges because of the inherent errors introduced by the quantization method. Nevertheless, results from Figure 3 indicate our method is not sensitive to the alternating step $T$ and is able to consistently enhance downstream fine-tuning performance.

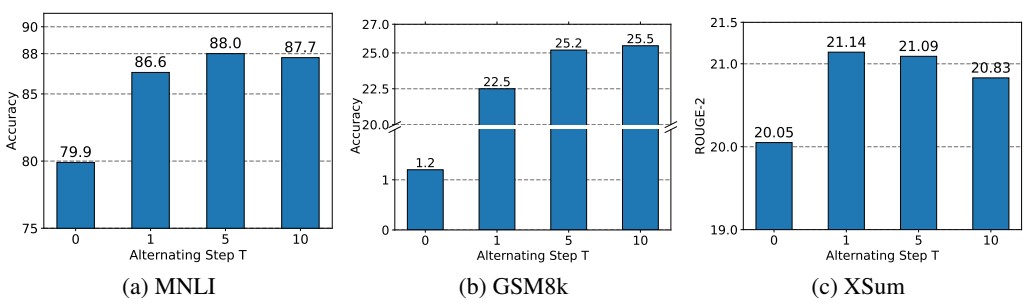

Figure 3: Comparison of different alternating step $T$ used in LoftQ. $T = 0$ indicates we use QLoRA method that initializes low-rank adapters by (5). $T = 1, 5, 10$ indicates we use different $T$ for LoftQ described in Algorithm 1. **Left**: Uniform 2-bit DeBERTaV3-base. **Middle**: NF2 2-bit LLAMA-2-13b. **Right**: NF4 BART-large.

## 5    DISCUSSION

**Start with quantization or SVD in the alternating optimization?** An alternative algorithm to the alternating optimization is that we first obtain the low-rank approximation $A_t, B_t$ and then obtain the quantized weight $Q_t$ by switching Line 3 and Line 4 in Algorithm 1. We note this is a valid alternative method as both still jointly minimize the objective in (6). Table 6 summarizes the performance of this alternative method. It is noteworthy that the alternative method still outperforms QLoRA significantly, even though it is worse than the primary version. This observation underscores the potential for performance improvement by achieving a closer approximation of pre-trained weights within the low-precision regime.

**LoftQ better than Full-precision LoRA?** We find LoftQ outperforms full precision LoRA in XSum and GSM8K (see Table 3 and Table 5). Beside the overfitting caused by lack of regularization, anonther possible explanation for this unexpected phenomenon is that the initial low-rank adapters obtained by LoftQ are non-zero while they are all zero in full precision LoRA as described in (5). Such zero initialization could make the fine-tuning unstable, and therefore it performs worse than LoftQ. We leave the study of the robustness of LoftQ as future work.

Table 6: Results of 2-bit uniformly quantized DeBERTaV3-base on part of GLUE. LoftQ(SVD First) indicates the alternative LoftQ that swiches Line 3 and Line 4 in Algorithm 1. We report the median over four random seeds. The best results on each task are shown in **bold**.

| Method | Rank | MNLI
m / mm | QNLI
Acc | SST2
Acc |
|---|---|---|---|---|
| Full FT | - | 90.5/90.6 | 94.0 | 95.3 |
| QLoRA | 32 | 79.9/79.5 | 83.8 | 86.6 |
| LoftQ(SVD First) | 32 | 87.8/87.7 | 84.9 | 89.7 |
| LoftQ(Quantiztion First) | 32 | **88.0/88.1** | **92.2** | **94.7** |

## 6    RELATED WORK

**Quantization-Aware Training (QAT)** is often used to obtain quantized models that are adapted in downstream tasks (Peri et al., 2020; Liu et al., 2023). It involves quantization and full model fine-tuning at the same time. However, QAT requires massive training cost, such as the gradient and optimization state. Moreover, it is difficult to compute the gradient of quantized weights. Our method, with the help of LoRA, sidesteps the aforementioned issues, providing a light approach for downstream task adaptation.

**Post-Training Quantization (PTQ)** is a category of popular quantization frameworks (Frantar et al., 2022; Xiao et al., 2023), which can also be used for task adaptation. It calibrates the high-precision model with a small subset of the training dataset. Therefore, the subsequent quantization is guided by the training dataset, providing task-specific quantized models. Besides, it does not involve any gradient backpropagation, so it is cost-efficient. However, it usually results in lower accuracy compared to QAT.

## 7    CONCLUSION

We propose LoftQ, a quantization framework for LLMs, which alternatively applies quantization and low-rank approximation to the original high-precision pre-trained weights, to obtain an initialization for the subsequent LoRA fine-tuning. Experiments on natural language understanding, question answering, summarization, and natural language generation show that our framework remarkably surpasses existing methods, e.g., QLoRA, for quantizing encoder-only, encoder-decoder, and decoder-only models. We have not observed our method exhibiting worse performance over QLoRA. Moreover, our quantization framework demonstrates effectiveness and robustness particularly in low-bit quantization regimes, e.g., the 2-bit level.

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

## A    MODEL COMPRESSION RATIO AND MEMORY FOOTPRINT

We report the compression ratio after applying LoftQ in Table 7. It is defined as

$$\text{compression ration} = \frac{\text{backbone size} + \text{LoRA adapter size}}{\text{pre-trained size}}.$$

We also measure the GPU memory cost during training. Given that GPU memory varies by models, tasks, sequence lengths, batch sizes, etc. We report LLAMA-2 on GSM8K as an example in Table 8.

Table 7: Compression ratios of backbones.

| Model | Compression ratio (%) | Trainable ratio (%) | Rank | Bits | Quantization method |
|---|---|---|---|---|---|
| DeBERTaV3-base | 15.6 | 3.1 | 16 | 2 | Uniform |
| DeBERTaV3-base | 18.8 | 6.3 | 32 | 2 | Uniform |
| DeBERTaV3-base | 17.2 | 3.1 | 16 | 2 | NF2 |
| DeBERTaV3-base | 20.4 | 6.3 | 32 | 2 | NF2 |
| BART-large | 15.3 | 1.2 | 8 | 4 | NF2 |
| BART-large | 16.7 | 2.5 | 16 | 4 | NF2 |
| BART-large | 27.8 | 1.2 | 8 | 4 | NF4 |
| BART-large | 29.0 | 2.5 | 16 | 4 | NF4 |
| BART-large | 26.2 | 1.2 | 8 | 4 | Uniform |
| BART-large | 27.5 | 2.5 | 16 | 4 | Uniform |
| LLAMA-2-7b | 16.6 | 2.4 | 64 | 2 | Nf2 |
| LLAMA-2-7b | 29.0 | 2.4 | 64 | 4 | Nf4 |
| LLAMA-2-13b | 16.0 | 1.9 | 64 | 2 | Nf2 |
| LLAMA-2-13b | 28.5 | 1.9 | 64 | 4 | Nf4 |

Table 8: GPU memory footprint

| Model | Dataset | Seq length | Batch size | GPU Mem |
|---|---|---|---|---|
| LLAMA-2-7b | GSM8K | 384 | 1 | 15GB |
| LLAMA-2-13b | GSM8K | 384 | 1 | 24GB |

## B    QUANTIZATION TIME

We report the execution time of LoftQ applying to a single weight matrix in Table 9. The time is tested on Intel(R) Xeon(R) CPU E5-2650 v4 @ 2.20GHz.

Table 9: Execution time of LoftQ applying to different weight matrices.

| Model | Size | Step $T$ | Quantization method | Time |
|---|---|---|---|---|
| DeBERTaV3-base | $768 \times 768$ | 5 | Uniform | 1s |
| BART-large | $1024 \times 1024$ | 5 | NF4 | 1s |
| LLAMA-2-7b | $4096 \times 4096$ | 5 | NF4 | 21s |
| LLAMA-2-13b | $5120 \times 5120$ | 5 | NF4 | 43s |

## C    GLUE DATASET STATISTICS

We present the dataset statistics of GLUE Wang et al. (2019) in the following table.

GLUE includes two single-sentence classification tasks: SST-2 (Socher et al., 2013) and CoLA (Warstadt et al., 2019), and three similarity and paraphrase tasks: MRPC (Dolan & Brockett, 2005), STS-B (Cer et al., 2017), and QQP. GLUE also includes four natural language inference tasks in GLUE: MNLI (Williams et al., 2018), QNLI (Rajpurkar et al., 2016), RTE (Dagan et al., 2007; Bar-Haim et al., 2006; Giampiccolo et al., 2007; Bentivogli et al., 2009), and WNLI (Levesque et al., 2012).

| Corpus | Task | #Train | #Dev | #Test | #Label | Metrics |
|--------|------|--------|------|-------|--------|---------|
| | | Single-Sentence Classification (GLUE) | | | | |
| CoLA | Acceptability | 8.5k | 1k | 1k | 2 | Matthews corr |
| SST | Sentiment | 67k | 872 | 1.8k | 2 | Accuracy |
| | | Pairwise Text Classification (GLUE) | | | | |
| MNLI | NLI | 393k | 20k | 20k | 3 | Accuracy |
| RTE | NLI | 2.5k | 276 | 3k | 2 | Accuracy |
| QQP | Paraphrase | 364k | 40k | 391k | 2 | Accuracy/F1 |
| MRPC | Paraphrase | 3.7k | 408 | 1.7k | 2 | Accuracy/F1 |
| QNLI | QA/NLI | 108k | 5.7k | 5.7k | 2 | Accuracy |
| | | Text Similarity (GLUE) | | | | |
| STS-B | Similarity | 7k | 1.5k | 1.4k | 1 | Pearson/Spearman corr |

Table 10: Summary of the GLUE benchmark.

## D  NATURAL LANGUAGE UNDERSTANDING

### D.1  GLUE WITH 4-BIT

We show the 4-bits results in the Table 11. Both methods can achieve performance close to full-finetuning.

Table 11: Results with 4-bit LoftQ of DeBERTaV3-base models on GLUE development set using NF4 quantization. We report the median over four seeds. Results with N.A. indicate the model does not converge. The best results on each dataset are shown in bold

| Method | Rank | MNLI m / mm | SST-2 Acc | QNLI Acc | ANLI Acc |
|--------|------|-------------|-----------|----------|----------|
| Full FT | - | 90.5/90.6 | 95.3 | 94.0 | 59.8 |
| QLoRA | 32 | 89.9/89.9 | **95.3** | **94.2** | 59.4 |
| LoftQ | 32 | **89.9/90.0** | 95.3 | 94.1 | **59.9** |

### D.2  TRAINING DETAILS

**Implementation Details.** The implementation of LoftQ is based on publicly available Huggingface (Paszke et al., 2019) code-base [3].

**Hyper-parameter Details.** We select the learning rate of $\{1 \times 10^{-5}, 5 \times 10^{-5}, 1 \times 10^{-4}, 5 \times 10^{-4}\}$, and use the selected learning rate for both uniform quantization experiments and nf2 quantization experiments. We use batch size of 32 for all GLUE tasks and ANLI. We use batch size of 16 for SQuADv1.1. We use LoftQ of 5 iterations for all GLUE tasks.

Table 12 summarizes the detailed hyperparameters for each task used in training DeBERTaV3-base using uniform quantization. Table 13 summarizes the detailed hyperparameters for each task used in training DeBERTaV3-base using nf2 quantization.

Table 12: Hyper-parameter setup of LoftQ for GLUE benchmark for training DeBERTaV3-base using Uniform quantization.

| Hyper-parameter | MNLI | RTE | QNLI | MRPC | QQP | SST-2 | CoLA | STS-B | SQuADv1.1 | ANLI |
|-----------------|------|-----|------|------|-----|-------|------|-------|-----------|------|
| # epochs | 5 | 20 | 10 | 60 | 10 | 10 | 60 | 60 | 10 | 12 |
| Learning rate | $1 \times 10^{-4}$ | $5 \times 10^{-4}$ | $5 \times 10^{-5}$ | $1 \times 10^{-4}$ | $5 \times 10^{-5}$ | $5 \times 10^{-5}$ | $5 \times 10^{-5}$ | $5 \times 10^{-5}$ | $5 \times 10^{-5}$ | $5 \times 10^{-5}$ |

---

[3] https://github.com/huggingface/transformers/tree/main/examples/pytorch

Table 13: Hyper-parameter setup of LoftQ for GLUE benchmark for training DeBERTaV3-base using NF2 quantization.

| Hyper-parameter | MNLI | RTE | QNLI | MRPC | QQP | SST-2 | CoLA | STS-B | SQuADv1.1 | ANLI |
|---|---|---|---|---|---|---|---|---|---|---|
| # epochs | 5 | 20 | 10 | 60 | 10 | 10 | 60 | 60 | 10 | 12 |
| Learning rate | $1 \times 10^{-4}$ | $5 \times 10^{-5}$ | $5 \times 10^{-5}$ | $1 \times 10^{-4}$ | $5 \times 10^{-5}$ | $5 \times 10^{-5}$ | $5 \times 10^{-5}$ | $1 \times 10^{-4}$ | $5 \times 10^{-5}$ | $5 \times 10^{-5}$ |

## E  Summarization

### E.1  Training Details

We choose Adam as the optimizer and try learning rate from$\{1 \times 10^{-5}, 5 \times 10^{-5}, 7 \times 10^{-5}, 2 \times 10^{-4}, 3 \times 10^{-4}, 4 \times 10^{-4}\}$. We show the optimal learning rate for different settings in Table 14. We use LoftQ of 1 iteration for all BART-large experiments. Table 14 and Table 15 summarize the learning rate and other hyper-parameters for CNN/DailyMail and XSum.

Table 14: Hyper-parameter setup of LoftQ BART-large on CNN/DailyMail

| Hyperparameter | NF4 | | 4-bit Uniform | | NF2 | |
|---|---|---|---|---|---|---|
| | rank8 | rank16 | rank8 | rank16 | rank8 | rank16 |
| Learning rate | 2e-4 | 2e-4 | 2e-4 | 3e-4 | 2e-4 | 2e-4 |
| Epoch | 15 | 15 | 15 | 15 | 15 | 15 |
| Batch size | 64 | 64 | 64 | 64 | 64 | 64 |

Table 15: Hyper-parameter setup of LoftQ BART-large on XSum

| Hyperparameter | NF4 | | 4-bit Uniform | | NF2 | |
|---|---|---|---|---|---|---|
| | rank8 | rank16 | rank8 | rank16 | rank8 | rank16 |
| Learning rate | 2e-4 | 2e-4 | 2e-4 | 2e-4 | 2e-4 | 2e-4 |
| Epoch | 25 | 25 | 25 | 25 | 25 | 25 |
| Batch size | 32 | 32 | 32 | 32 | 32 | 32 |

## F  Natural Language Generation

We set the batch size as 32 for WikiText-2 and 16 for GSM8K. We train 2 epochs on WikiText-2 and 6 epochs on GSM8K. We select learning rate from$\{1 \times 10^{-5}, 5 \times 10^{-5}, 7 \times 10^{-5}, 1 \times 10^{-4}, , 3 \times 10^{-4}, 4 \times 10^{-4}\}$. Specific settings are summarized in Table 16 and Table 17.

## G  Comparison to Pruning

Pruning is also a widely used compression method. Here we compare LoftQ with the state-of-the-art pruning method Li et al. (2023). We show the comparison in Table 18. We can see our method significantly outperforms the pruning methods on DeBERTaV3-base model. We also remark that LoftQ can consistently reduce the memory of both training and storage. In contrast, pruning requires training the entire full-precision matrix, which implies that it can not achieve any memory savings during the training stage.

## H  Extension to Convolutional Layers

Low-rank adapters can also be applied to convolutional layers. Given an input feature map $X \in \mathbb{R}^{h \times w \times c_1}$ and $c_2$ 2D convolutional kernels $K_i \in \mathbb{R}^{c_1 \times d \times d}, i = 1, 2, ..., c_2$, the output of the convolutional layer is

$$Y = \text{stack}(X \otimes K_1, ..., X \otimes K_{c_2}), \tag{10}$$

where $Y \in \mathbb{R}^{h \times w \times c_2}$ and $\otimes$ denotes the 2D convolution operation.

Table 16: Hyper-parameter setup of LoftQ LLAMA-2-series on GSM8K

| Model | Hyperparameter | NF4 | NF2 | Mixed-precision |
|---|---|---|---|---|
| LLAMA-2-7b | learning rate | $3 \times 10^{-4}$ | $3 \times 10^{-4}$ | $3 \times 10^{-4}$ |
| LLAMA-2-13b | learning rate | $1 \times 10^{-4}$ | $1 \times 10^{-4}$ | $3 \times 10^{-4}$ |

Table 17: Hyper-parameter setup of LoftQ LLAMA-2-series on WikiText-2

| Model | Hyperparameter | NF4 | NF2 | Mixed-precision |
|---|---|---|---|---|
| LLAMA-2-7b | learning rate | $3 \times 10^{-4}$ | $3 \times 10^{-4}$ | $3 \times 10^{-4}$ |
| LLAMA-2-13b | learning rate | $1 \times 10^{-4}$ | $1 \times 10^{-4}$ | $3 \times 10^{-4}$ |

Table 18: Results of LoftQ using 2-bits uniform quantization compared with LoSparse with DeBERTaV3-base models on some of GLUE development sets. Here *Ratio* is the proportion of total remaining weights. Results with *N.A.* indicate the model does not converge.

| Method | Ratio | MNLI
m / mm | SST-2
Acc | QNLI
Acc |
|---|---|---|---|---|
| Full FT | 100% | 90.5 / 90.6 | 95.3 | 94.0 |
| LoSparse | 15% | 83.3/82.9 | 87.6 | 90.4 |
| | 20% | 84.5/83.8 | 91.7 | 88.6 |
| LoftQ | 15.6% | **87.3/87.1** | **94.0** | **90.6** |
| | 18.8% | **88.0/88.1** | **94.7** | **92.4** |

We can reformulate Equation (10) into matrix multiplication as

$$Y = Z \times H^\top,$$

where $Z \in \mathbb{R}^{hw \times c_1 d^2}, H \in \mathbb{R}^{c_2 \times c_1 d^2}$, by extending and flattening the input $X$ together with concatenating and flattening kernels. We first extend a vector $x_{i,j} \in \mathbb{R}^{c_1}$ by its neighbor vectors within the kernel window:

$$x'_{i,j} = \text{Concat}(\mathrm{x}_{i-\frac{d}{2},j-\frac{d}{2}}, ..., \mathrm{x}_{i+\frac{d}{2},j+\frac{d}{2}}).$$

Now, $X$ becomes $X' \in \mathbb{R}^{h \times w \times c_1 d^2}$. We then flatten $X'$ into $Z \in \mathbb{R}^{hw \times c_1 d^2}$. For kernels, we first concatenate $\{K_1, ..., K_{c_2}\}$ into $H' \in \mathbb{R}^{c_2 \times c_1 \times d \times d}$. We then flatten $H'$ into $H$.

Note that $H$ can be approximated by a low-rank matrix

$$R = UV^\top,$$

where $U \in \mathbb{R}^{c_2 \times r}, V \in \mathbb{R}^{c_1 d^2 \times r}, r \ll \min\{c_2, c_1 d^2\}$ by SVD. Therefore, the original convolution layer can be approximated as

$$\widehat{Y} = Z \times (UV^\top)^\top \tag{11}$$

$$= (Z \times V) \times U^\top \tag{12}$$

$$= M \times U^\top. \tag{13}$$

Note that $Z \times V$ can be restored into a convolution operation where we have $r$ kernels $D_i \in \mathbb{R}^{c_1 \times d \times d}, i = 1, 2, , ..., r$ and $M \times U^\top$ can also be restored into a convolution operation where we have $c_2$ kernels $U_i \in \mathbb{R}^{r \times 1 \times 1}, i = 1, 2, , ..., c_2$.

