# OpenReview forum: "LoftQ: LoRA-Fine-Tuning-aware Quantization for Large Language Models"
_ICLR.cc/2024/Conference — ICLR 2024 oral_

### Official Review · Reviewer_BzhB · 2023-11-01

**Soundness:** 3 good
**Presentation:** 3 good
**Contribution:** 3 good
**Rating:** 8
**Confidence:** 4

**Summary:**

This paper introduces a new approach for weight quantisation and parameter-efficient fine-tuning via low-rank adapters termed LoftQ. LoftQ is inspired by QLoRA and aims to improve over it by providing a better quantisation and better initialisation for the low-rank adapter weight matrices.

For background: LoRA makes the assumption that the difference between pre-trained and fine-tuned weights can be approximated by a low-rank matrix, i.e. $W_{ft*} = W_{pt} + AB^T$.

The core contribution of this work relies on the observation that QLoRA quantises $W_{pt}$ but still relies on the default LoRA initialisation which assumes a non-quantised matrix $W_{pt}$.

To address this shortcoming, the authors propose an iterative LoRA-aware quantisation which jointly improves the quantisation of $W_{pt}$, making it more similar to the pre-trained weight, and the initialisation of $A$ and $B$ (as the authors note, QLoRA is a special case of their proposed algorithm).

The authors compare their proposed approach to QLoRA and full fine-tuning across several models and datasets, showing that it consistently outperforms QLoRA.

In addition to their main experiments, the authors provide ablations investigating their proposed approach in more detail.

- Dettmers et al. 2023 - QLoRA: Efficient Finetuning of Quantized LLMs

**Strengths:**

- The core contribution of this work is well motivated and grounded in the shortcomings of an existing widely used approach.
- The authors provide sufficient experimental results to demonstrate the usefulness of their approach
- The authors provide ablation studies, investigating important details of their approach
- The paper is well written, the structure is clear and easy to follow

**Weaknesses:**

I couldn't identify serious weaknesses of this work but I have some suggestions and questions for the authors. See below.

**Questions:**

**Questions and suggestions**
- The result of the LoftQ algorithm is a quantised weight matrix ($Q_T$) as well as the LoRA matrices ($A_T$, $B_T$). An interesting ablation would be to discard $A_T$ and $B_T$ and use the default LoRA initialisation instead. This would tell us more about the importance of initialising $A_T$ and $B_T$ differently.
- One of the findings in the QLoRA paper is that it is crucial to add LoRA adapters to every linear layer of the model (Figure 2 in the QLoRA paper). It could be interesting to run a similar ablation with your method. Given your improved initialisation, maybe it is sufficient to add LoRA adapters to fewer layers.
- It could be interesting to study the difference in initialisation of the low-rank matrices more. Does your work provide insights into what makes a good LoRA initialisation and could these insights be potentially applied to non-quantised LoRA as well?


**Typos and writing suggestions**

- Introduction, second paragraph: "It is predicated on the hypothesis ..."
    - You might want to use "based" instead of predicated
- Discussion, LoftQ better than full precision LoRA: "Such zero initialisation could cause the fine-tuning unstable"
    - This sentence needs rewriting

---

> ### Author Response · Authors · 2023-11-22
> **Response to the questions**
>
> Thank you for pointing our typos and writing suggestions, we will improve it in the next version.
>
> **Response to the questions**
>
> 1. LoftQ enables $Q_T + A_T B_T$ as close to the pre-trained weight $W$ as possible. It does not guarantee $Q_T$ alone would be a better initialization than $Q_0$ which is used by QLoRA. We verify this in two ways: (1) we compare the F norm $||W-Q_T||_F$ and $||W-Q_0||_F$; (2) we take experiments on $Q_T$ with default LoRA initialization, as you suggest. The results are listed below:
>
> (1). We measure the fc1 matrix of the 1st encoder layer in BART-large.
>
> |                         | t=0  | t=1  | t=2  | t=3  | t=4  |
> | -------------------------------- | ---- | ---- | ---- | ---- | ---- |
> | $ \|\|W - Q_t\|\|_F$                  | 10.50 | 10.69 | 10.99 | 11.28 | 11.55 |
> | $ \|\|W-(Q_t + A_t B_t^{\top})\|\|_F$ | 10.05   | 9.73   | 9.56   | 9.44  | 9.36   |
>
> From the table, we can see  $||W - Q_t||_F$ actually increases throughout more iterations while $||W - (Q_t + L_t R_t^{\top})||_F$ decreases.
>
> (2). We also take experiments on GSM8K
> |    Model    | $Q_T + A_T B_T$ |  $Q_T$ w/ zero init  |
> | ----------- | --------------  | ----------- |
> | LLAMA-2-7b  |      35.0       |     33.4    |
> | LLAMA-2-13b |      45.0       |     41.6    |
>
> 2. This is an interesting question, but it is beyond the scope of our work. It remains open on how to achieve the best trade off between the number of layers with LoRA adapters and the performance. We would suggest referring to [AdaLoRA](https://arxiv.org/abs/2303.10512), which allocates dynamic ranks to different frozen matrices.
>
> 3. The insight of LoftQ is that the full low-rank adapters are used to bridge the gap brought by quantizing pre-trained weights. We, unfortunately, cannot apply this logic to non-quantized LoRA because it does not have quantization discrepancy issues.

---

### Official Review · Reviewer_5E8o · 2023-11-10

**Soundness:** 4 excellent
**Presentation:** 4 excellent
**Contribution:** 4 excellent
**Rating:** 8
**Confidence:** 4

**Summary:**

This paper introduces a method (LoftQ) to initialize the quantized weights in a transformer based model for future LoRA-based fine-tuning. Different from initializations for Quantized Lora used in prior methods, such as fixup or zero-init, LoftQ initalizes the quantized matrix weights and lora weights together to minimize the Frobenious norm of the difference between the floating point weights and the quantized weights. The initialization process is iterative where the quantized matrix is obtained through a standarized quantization process and the lora quantized weights are obtained from a SVD decomposition.

Experiments on encoder models (classification), encoder-decoder models (summarization), and decoder models (math reasoning, language modeling) are conducted and results are in favor of the LoftQ initialization.

**Strengths:**

1. The lack of a proper initialization of quanitzed lora methods intuitively makes sense, the authors identified this problem and proposed a simple but working solution to address this problem. I appreciate this simplicity.
2. The experiments are well conducted over quite a few domains/datasets, models, and quantization schemas.
3. The paper is well written.

**Weaknesses:**

1. It might be better to put higher priority and conduct more experiments on decoder-based (or encoder decoder) models for generative tasks. It seems that quantized lora (whether with or without intialization) lacks too much in classification tasks with encoders, to the extent that pratictionars probably won't want to train quantized lora models on these tasks.
2. Otherwise, I find this paper well rounded without significant weaknesses.

**Questions:**

1. It would be nice to show the memory footprint for 2-bit quantized models during training.
2. Would the quantized lora initialization in turn help full quantized fine-tuning?

---

> ### Author Response · Authors · 2023-11-22
> **Response to the weaknesses and questions**
>
> **Response to the weaknesses**
>
> We appreciate your suggestion on conducting more experiments on decoder-only models. We are currently working on LLAMA-2-70b. Due to the time limit, we may be not able to present comprehensive results in rebuttal within a short response period. However, we will add more experiments in the next version.
>
> **Response to the questions**
>
> 1. The training memory footprint varies by task, batch size, and input sequence length. Therefore, we only provide examples.
> | Model       | Bit  | Rank | Task  | Seq length | Batch Size | GPU Memory |
> | ----------- | ---- | ---- | ----- | ---------- | ---------- | ---------- |
> | LLAMA-2-7b  | 4    | 64   | GSM8K | 384        | 1         | 15GB       |
> | LLAMA-2-13b | 4    | 64   | GSM8K | 384        | 1         |  24GB   |
>
>
> 2. We are not sure about what you mean by “full quantized fine-tuning”. If it means quantization-aware full fine-tuning, our method is orthogonal to this setting because quantization-aware full fine-tuning does not have low-rank adapters. However, in terms of the initialization of the quantization-aware full fine-tuning, which also starts with a quantized model, our method could provide valuable insight: simply add the $A_T B_T$ back to $Q_T$, but how to add it back so that it achieves the least quantization discrepancy still remains an open question.

---

### Official Review · Reviewer_tN6u · 2023-11-11

**Soundness:** 4 excellent
**Presentation:** 4 excellent
**Contribution:** 3 good
**Rating:** 6
**Confidence:** 4

**Summary:**

This work proposes better initialization for LoRA adaptors A and B, and the Quantization of pre-trained weights W_{pt} in a setup where two things are desired:
1) downstream fine-tuning
2) quantization of W_{pt}.

The authors propose an iterative method to find better initializations for these matrices. Through rigorous experiments the work shows that the proposed initialization is better than the vanilla initialization proposed in QLoRA.
The authors conduct experiments with almost-extreme quantization (2 bit) to show efficacy of their approach, where the traditional methods (QLoRA) even fail to train.
The work also attempts to analyze the impact of number iterations (of the proposed iterative method) and the experiments are conducted well.

**Strengths:**

* This work presents well motivated initialization method for LoRA + Quantization
* Through extensive experimentation on several architectures and benchmarks, this work clearly elucidates pitfalls of QLoRA and effectiveness of the proposed method

**Weaknesses:**

None, but a few clarifying questions stated below.

**Questions:**

1) For the XSUM and GSM8k tasks, LoftQ gets better accuracy than full-precision LoRA. I wonder how the FP LoRA was tuned? Maybe 4 bit quantization does implicit regularization, and FP LoRA  was not regularized well enough? This would especially make a difference if the tasks are low dimensional. In other words, if a high capacity LLAMA 13B model is fine-tuned LoRA style on GSM8k, how did the authors ensure that the model was not overfitted?

2) It would be nice to analyze number of epochs, and training steps required for baseline full precision LoRA and LoftQ.

3) LoRA's original motivation stems from "training efficiency" while maintaining the inference cost the same as the base model. Conversely quantization's main motivation is inference efficiency. Keeping training efficiency aside, a good baseline maybe quantization aware fine-tuning (i.e. no LoRA), to establish upper bound on accuracy for LoftQ.

4) It wasn't very fully clear but are the LoRA adaptors, A and B, quantized as well in LoftQ?

---

> ### Author Response · Authors · 2023-11-22
>
> **Response to your questions**
> 1. Thanks for the advice of taking experiments on regularized FP LoRA. The FP LoRA in our paper is fine-tuned without any regularization but we have properly tuned the hyper-parameters, including lora_alpha, learning rate, batch size, etc. We did observe overfitting: as the training epoch increases, the accuracy on the test set drops after a certain epoch. We use early stopping to prevent overfitting. We stop the training by the 6th epoch.
>
> 2. Adding weight decay to the LoRA adapters does help the LLAMA-2-13b on GSM8K, but it does not help LLAMA-2-7b on GSM8K. Specifically, full precision LLAMA-2-13b with LoRA achieves 46.0% now, instead of 43%, but LLAMA-2-7b drops from 36.9% to 34.4% on GSM8K. We will add the regularization results to the latest paper.
> We have listed the required epochs in Table 11 and Table 12 in the Appendix. We choose the same epochs for both FP LoRA and LoftQ in all experiments.
> Thank you for your insightful comments. Allow me to address a potential misunderstanding regarding the motivation of LoRA and quantization. The original motivation of LoRA is efficient multi-task learning as introduced in the second paragraph of Section 1 in LoRA paper. The motivation of weights-only quantization is model compression, allowing LLMs to run on memory-limited devices.
>
> 3. To the best of our knowledge and the NLP literature, it is not clear whether quantization-aware training (QAT) would be an upper bound for LoftQ or QLoRA, given two reasons: (1) the initial weights are quantized and therefore there exists a significant gap between quantized initialization and the pre-trained weights in QAT, while LoftQ can reduce such a gap. (2) it is difficult to update the non-continuous quantized parameters, which is merely possible to achieve full precision fine-tuning performance, let alone given severely degraded initial weights. On the other hand, LoftQ fine-tunes full precision LoRA adapters, making it easy to fine-tune.
> The LoRA adapters, A and B, are not quantized. They remain full precision throughout the training. Maintaining full precision or for the LoRA adapters cause minor extra memory compared to quantized LoRA adapters.

---

### Public Comment · ~Baohao_Liao1 · 2023-11-21
**Attention to a closely related work**

Dear authors,

I enjoy reading this paper, since it is well-motivated, well-written, and offers strong results. In this context, I would like to draw your attention to our closely related paper published in NeurIPS2023 (https://openreview.net/forum?id=J8McuwS3zY) [1]. I see some similarities in the motivation to preserve the starting point from the pre-trained model. Please consider including our work in your section of "Related Work" as I believe it is indeed a closely related work.

The main similarities are:
1. In our paper, we thoroughly investigate why it is important to preserve the starting point from the pre-trained model (Section 2.2) and propose the starting point hypothesis. In your paper, you are also motivated by preserving the starting point, initializing LoRA in a way to make the representation similar between before and after quantization.
2. Both works focus on memory-efficient fine-tuning. Your work improves QLoRA and our work modifies a pre-trained model to make it reversible.


[1] Make Pre-trained Model Reversible: From Parameter to Memory Efficient Fine-Tuning, Baohao Liao, Shaomu Tan, Christof Monz

---

### Meta-Review · Area_Chair_4nEc · 2023-12-14

**Metareview:**

This paper introduces a method (LoftQ) to initialize the quantized weights in a transformer based model for future LoRA-based fine-tuning. LoftQ initializes the quantized matrix weights and lora weights together to minimize the Frobenius norm of the difference between the floating point weights and the quantized weights.
The paper is well written and there are no weaknesses raised by the reviewers. The paper is well motivated and grounded in the shortcomings of an existing widely used approach. The experiments are well conducted over a large number of datasets, models, and quantization schemas. The paper provides sufficient experimental results to demonstrate the usefulness of their approach.

**Justification For Why Not Higher Score:**

N/A

**Justification For Why Not Lower Score:**

* Important, relevant topic with extensive experimental results
* Not a single weakness raised by the reviewers
* Potential high impact publication

---

### Decision · Program_Chairs · 2024-01-16

Accept (oral)